# PAX6 Downregulation Triggers HIF-1α-Mediated Ferroptosis in Glioma Cells

**DOI:** 10.3390/biom15101462

**Published:** 2025-10-16

**Authors:** Qizhi Luo, Li Fu, Jie Zhang, Shashuang Zhang, Lixiang Wu, Quan Zhu, Baisheng Huang

**Affiliations:** 1Experimental Center for Cellular and Molecular Biology, School of Basic Medical Sciences, Central South University, Changsha 410017, China; 802611@csu.edu.cn (Q.L.); 803164@csu.edu.cn (J.Z.); 220263@csu.edu.cn (S.Z.); 2Department of Immunology, School of Basic Medical Sciences, Central South University, Changsha 410017, China; 246511060@csu.edu.cn; 3Department of Physiology, School of Basic Medical Sciences, Central South University, Changsha 410017, China; ywlx@csu.edu.cn; 4Department of Immunology, Medical College, Hunan University of Chinese Medicine, Changsha 410208, China; 5Top-Notch Innovation Base of Basic Medicine, Central South University, Changsha 410017, China

**Keywords:** PAX6, HIF-1α, glioma, ferroptosis

## Abstract

**Background**: The paired-box gene 6 (*PAX6*) is an important transcription factor in the central nervous system, mainly regulating the development and differentiation of embryonic eyes and the nervous system. PAX6 expression is significantly decreased in glioma, and the expression levels are closely related to glioma development and prognosis. Therefore, it is important to study and elucidate the biological function of PAX6 in glioma to further our understanding of the occurrence and development of glioma. **Methods**: This study focused on the expression and regulation of PAX6 and hypoxia-inducing factor (HIF-1α) and investigated the molecular mechanism of ferroptosis regulated by PAX6 and HIF-1α. Firstly, immunohistochemistry, qPCR, Western blot, and other methods were used to detect PAX6 and HIF-1α expression in glioma tissues and cells, as well as the specific way in which PAX6 regulates HIF-1α. Then, some relative indicators of ferroptosis regulated by PAX6 in glioma were studied. **Results**: The results showed that PAX6 inhibited HIF-1α expression by regulating the levels of reactive oxygen species (ROS); overexpression of PAX6 promoted the expression of ROS and lipid peroxides (LPOs) in glioma cells and decreased the expression of intracellular antioxidant systems glutathione peroxidase 4 (GPX4) and glutathione (GSH). **Conclusions**: Downregulation of PAX6 plays an important role in regulating ferroptosis in glioma cells. Our research provides a reference basis for a deeper understanding of the role of PAX6 in ferroptosis of glioma.

## 1. Introduction

Gliomas are the most common and highly malignant tumors of the central nervous system, with a five-year survival rate below 10% [1]. Due to their unique physiological and anatomical characteristics, gliomas exhibit poor responsiveness to conventional therapies such as radiotherapy and chemotherapy [2,3]. Their highly invasive growth also means that many patients are not eligible for radical surgical treatment at diagnosis [4]. The pathogenesis of gliomas is extremely intricate and remains incompletely understood, and it is often accompanied by disorders and imbalances in the expression profiles of numerous genes [5,6,7]. Exploring the biological functions of the relevant genes in gliomas will elucidate their molecular pathogenic mechanisms and may identify tumor suppressor targets with potential clinical translational value, providing new strategies for precisely treating gliomas.

The paired-box gene 6 (*PAX6*), a transcription factor, is known for its essential roles in embryonic development, ocular morphogenesis, and tumor progression [8]. PAX6 exhibits dual regulatory roles in cancer, with its dysregulated expression contributing to tumor development [9,10]. Studies have shown that PAX6 expression is decreased in gliomas and is positively correlated with a glioma patient’s prognosis. Overexpression of PAX6 can inhibit the proliferative capacity of glioma cells [11,12,13]. In contrast, PAX6 is overexpressed in non-small-cell lung cancer, where it mediates tumor cell proliferation, invasion, and cell cycle progression through the ERK and MAPK signaling pathways. While PAX6 is known to suppress glioma cell proliferation via mechanisms involving the Sonic hedgehog pathway, VEGF signaling, and the sphingosine kinase, its downstream molecular targets and regulatory network remain inadequately defined [14,15,16]. Our previous research demonstrated that PAX6 is regulated by microRNA-223, and the overexpression of PAX6 significantly inhibited glioma cell proliferation and invasion while increasing sensitivity to the chemotherapeutic agent temozolomide [17,18]. However, how PAX6 plays a tumor-suppressing role in glioma and the specific molecular basis of PAX6’s tumor-suppressive function in gliomas needs to be further elucidated.

Previous studies indicate that overexpression of PAX6 markedly reduces the protein level of hypoxia inducible factor-1α (HIF-1α) in glioma cells. As a key regulator of hypoxia response, it is frequently overexpressed in solid tumors, such as gliomas, and it is intricately associated with tumor metastasis and poor prognosis [19,20,21,22]. The abnormal vascular proliferation characteristic of gliomas results in their core region often being under severe hypoxia. Under such conditions, HIF-1α can significantly promote cellular glycolysis via the Warburg effect to support cellular adaptation to the hypoxic environment [23]. The elevated production of HIF-1α is a significant factor enabling tumor cells to endure in hypoxic or even anaerobic conditions and proliferate rapidly. HIF-1α has also been implicated in multiple signaling pathways that mediate glioma growth, chemotherapy resistance, and immune escape, including involvement in the PI3K/Akt/mTOR pathway to enhance cell survival, inducing ERK phosphorylation to promote tumor cell invasion, and regulating JAK2/STAT3 to increase the expression of immune checkpoints, etc. [24,25,26,27,28,29]. These findings suggest that HIF-1α functions as an oncogene in glioma.

Although a negative correlation between PAX6 and HIF-1α has been proposed, a direct molecular mechanism of interaction between PAX6 and HIF-1α has not been demonstrated. Chen et al. discovered that the upregulation of PAX6 expression during mouse embryonic eye development preserved the normal development of mouse lens morphology, while targeted knockdown of the *HIF-1α* gene in embryonic lens epithelial cells also rectified the abnormal ocular developmental phenotype. This indicates a potential functional relationship between PAX6 and HIF-1α in ocular tissue, characterized by functional antagonism in regulating homeostasis [30]. In tumor studies, Pastorek found that ovarian cancer cells induced the expression of tumor-suppressing genes like PAX6 while suppressing pro-oncogenic genes such as HIF-1α following sulforaphane intervention in a hypoxic microenvironment, further suggesting that there may be a negative correlation at the level of transcriptional regulation between PAX6 and HIF-1α [31]. However, the above-mentioned study only speculated from the gene expression level that the expression of PAX6 and HIF-1α might be negatively correlated but failed to reflect whether PAX6 regulates HIF-1α; the biological correlation between the two also lacks relevant evidence. Consequently, it is imperative to investigate the influence of the regulatory interaction between PAX6 and HIF-1α in glioma formation and development. Recent studies indicate that HIF-1α promotes tumor cell survival by inhibiting ferroptosis. For instance, HIF-1α can improve the resistance of non-small-cell lung cancer to chemotherapeutic agents by inhibiting ferroptosis [32]. Additionally, Yang et al. discovered that inhibiting HIF-1α in oral squamous cell carcinoma elevates ROS levels and facilitates ferroptosis in tumor cells, marking it as a potential target for therapy [33]. These findings imply that HIF-1α may serve as a crucial regulator of ferroptosis, sustaining the malignant phenotype of tumor cells through the intricate modulation regulation of oxidative stress pathways.

In this study, we established glioma cell lines with stable PAX6 overexpression and integrated bioinformatics, molecular, and animal model approaches to systematically investigate the regulatory relationship between PAX6 and HIF-1α. Furthermore, we explored the potential mechanistic role of PAX6 dysregulation in modulating HIF-1α and ferroptosis in gliomas. Our findings highlight PAX6 as a pivotal regulator of ferroptosis, offering new insights into glioma pathophysiology and uncovering promising targets for therapeutic intervention.

## 2. Materials and Methods

### 2.1. Main Reagents

Dulbecco’s modified Eagle medium (DMEM) was purchased from Hyclone, Logan, UT, USA. Fetal bovine serum (FBS) was purchased from Gibco, New York, NY, USA. Total RNA extraction kit, 2× SYBR Green real-time quantitative real-time PCR (qPCR) premix, anti-PAX6 antibody, anti-HIF-1α antibody, anti-GPX4 antibody, anti-GAPDH antibody, and horseradish peroxidase (HRP)-labeled goat-anti-rabbit IgG were purchased from Cusabio, Shanghai, China. All the antibodies mentioned here are rabbit-derived polyclonal antibodies. The RIPA cell lysate and penicillin–streptomycin (P/S) mixture, reactive oxygen species (ROS) assay kit, lipid peroxide (LPO) assay kit, and glutathione (GSH) assay kit were purchased from Solarbio, Beijing, China. The ferrous ions (Fe^2+^) assay kit was purchased from Elabscience, Wuhan, China. Ferrostatin-1 (Fer-1) was purchased from MedChemExpress Company (MCE), South Brunswick, NJ, USA. All the plasmids included in our research, namely pLVX-IRES-ZsGreen-PAX6, pLVX-IRES-ZsGreen, psPAX2, pMD2.G, pcDNA3.1-PAX6, and pcDNA3.1-Con, were purchased from Changsha Kuaimai Company, Changsha, China.

### 2.2. Cell Culture and Processing

Glioma cell lines (U251) were purchased from the China Center for Type Culture Collection (CCTCC) of Wuhan University, HEK293 cells was purchased from the cell bank of the Chinese Academy of Sciences. The U373 cells and human microglia cells (HMO6) are kept by our laboratory. HEK293, U251, U373, and HMO6 cells were cultured in DMEM containing 10% FBS and 1% P/S at 37 °C with 5% CO_2_.

### 2.3. Immunohistochemical Analysis of Tissue Microarrays

The glioma tissue microarrays used in this study were purchased from Zhongke Guanghua (Xi’an, China) Intelligent Biotechnology Co., Ltd. (Bioaitech Co., Ltd., Hangzhou, China) and contained 11 normal brain tissues and 98 glioma tissue samples. Immunohistochemical staining was performed using anti-PAX6 antibody and anti-HIF-1α antibody produced by Millipore, Burlington, MA, USA. These two antibodies are both rabbit-derived monoclonal antibodies. After staining, the sections were analyzed by full-field imaging with the Pannoramic MIDI digital section scanning system from 3DHISTECH (iViewerSetup-7.2.5.1 version), Hungary. The results were interpreted according to the following criteria: no staining was negative (−), light brown color was weakly positive (+), brown color was positive (++), and tan color was strongly positive (+++).

### 2.4. Establishment of Stable Cell Lines

To investigate the effect of PAX6 on glioma cells, we established U251 and U373 stable cell lines overexpressing PAX6. First, HEK293 cells were cultured and supplemented with fresh DMEM during the logarithmic growth phase. At this point, the cells are proliferating exponentially and are in their optimal state. pLVX-IRES-ZsGreen-PAX6 or pLVX-IRES-ZsGreen plasmid with psPAX2 and pMD2.G plasmids were dissolved in serum-free, antibiotic-free DMEM at a mass ratio of 15:11:4. Then, 1 mg/mL of polyethyleneimine (PEI) was added and incubated at 25 °C for 15 min. Subsequently, the mixture was gently added to the HEK293 cells, and the medium was replaced after 6 h. The supernatant was collected after 48 or 72 h, and cellular debris was removed by passing it through a 0.45 μm filter, transferring it to U251/U373 cells in a 6-well plate, and the cationic carrier polybrene was added at a volume ratio of 1000:1. After centrifugation at 1600 rpm for 1 h, all cells were incubated for 48 h at 37 °C with 5% CO2. Fluorescent cells were counted using a fluorescence microscope (Leica, Wetzlar, Germany, DFC7000T), and 1 μg/mL of puromycin was added to screen for PAX6-expressing cells (PAX6-OE) or control cells (Con).

### 2.5. Quantitative Real-Time PCR (qPCR)

The supernatants of cultured U251, U373, and HMO6 cells were discarded and lysed by adding cell lysis solution containing 1% β-mercaptoethanol. Total cellular RNA was extracted according to the instructions of the RNA extraction kit (Ecotop, Guangzhou, China), and the RNA concentration was determined by using pre-cooled RNAase-free water to elute the total RNA bound on the adsorbent membrane. The above cell-derived RNA samples were diluted to 100 ng/μL, 1 μL of RNA was taken for reverse transcription by the cDNA reverse transcription kit (AG biology, Shanghai, China), and the remaining RNA was stored at −80 °C. A 20 μL reaction system was formulated as follows: 1 μL of cDNA template + 10 μL of 2× SYBR Green premixed solution + 2 μL of *PAX6/HIF-1α/GAPDH* primers + 7 μL enzyme-free water. The reaction program was as follows: initial denaturation at 95 °C for 1 min, followed by 40 cycles (denaturation at 95 °C for 30 s → annealing at 60 °C for 10 s → extension at 72 °C for 30 s). The Ct value was read, and the mRNA expression level of each gene was calculated by the relative expression formula N = 2^−ΔΔCt^. Each indicator was detected three times, and the average value was taken. The related primer sequences were as follows (Table 1).

### 2.6. Western Blot

U251, U373, and HMO6 cells were cultured for a specific period of time, 200 μL of RIPA lysis solution containing 1% phenylmethylsulfonyl fluoride (PMSF) was added to each well of the cells, and then the cells were lysed on ice for 30 min. The precipitate was discarded by centrifugation at 13,000 rpm for 10 min, and the supernatant was retained. The concentration of each group of proteins was determined according to the instructions of the BCA kit and quantified to 35 μg. After SDS-PAGE electrophoresis, the proteins were transferred to a PVDF membrane with a current of 200 mA. Subsequently, the membrane was blocked with 5% skim milk powder and incubated at 37 °C for 1 h. Monoclonal antibodies against specific antigens (PAX6, HIF-1α, GPX4, and GAPDH) were diluted proportionally in 5% skim milk powder and incubated at 4 °C overnight. On the following day, the membrane was washed 3 times with PBS buffer containing 0.1% Tween-20 (0.1% PBST), goat anti-rabbit IgG secondary antibody (1:4000 dilution) was added, and incubation was carried out at 37 °C for 1 h. After the membrane was washed 3 times with PBST, the membrane was developed and exposed by dropwise addition of ECL chemiluminescent solution.

### 2.7. CCK-8 Assay

Stable cell lines (U251/U373-PAX6-OE, U251/U373-Con) were seeded with inoculation in 96-well plates at a density of 5 × 10^4^ cells/mL, and 100 μL of cell suspension was added to each well. Subsequently, 10 μL of CCK-8 solution was added to each well, and after incubation at 37 °C for 4 h, the optical density (OD) values were measured at 450 nm using an enzyme marker. A total of 8 96-well plates were used, corresponding to the cell proliferation from day 0 to day 7. Any treatment conditions were repeated 5 times.

### 2.8. Lipid Peroxide (LPO) Detection

A lipid peroxide (LPO) detection kit was used to determine LPO levels in glioma U251 and U373 cells. The cells were collected after 48 h of culture, processed by sonication, and the experiments were carried out in strict accordance with the operating procedures of the kit instructions. The absorbance (OD value) was measured by using an enzyme immunoassay analyzer (AMR-100, Ausheng, Hangzhou, China) under an excitation light of 586 nm, and the LPO concentration was calculated according to the formula in the instruction manual. Before the test, we diluted the standard solution with a concentration of 1000 nmol/mL using the standard solution diluent (provided by reagents) to 20, 10, 5, 2.5, 1.25, 0.625, 0.3125, and 0.15625 nmol/mL. Each standard solution was measured three times. A standard curve was established based on the concentration (x, nmol/mL) of the standard tubes and the absorbance value (y, OD). Using the standard curve, the concentration of LPO in each sample was detected according to the standard curve. Any treatment conditions were repeated 3 times.

### 2.9. Reactive Oxygen Species (ROS) Detection

A reactive oxygen species (ROS) assay kit was used to determine ROS levels in U251 and U373 cells. The fluorescent probe dihydrofluorescein diacetate (DCFH-DA) was diluted at a ratio of 1:2000 and incubated with the cells at 37 °C and 5% CO_2_ for 20 min. Green fluorescent signals were then observed under an excitation light of 488 nm using a fluorescence microscope (Leica, Germany, DFC7000T), and fluorescence intensity was used to quantify the intracellular ROS levels.

### 2.10. Glutathione (GSH) Assay

A glutathione (GSH) assay kit was used to determine GSH levels in U251 and U373 cells. The cells were collected after 48 h of culture, processed by ultrasonic crushing, and the experiments were carried out in strict accordance with the operating procedures of the kit instructions. The absorbance (OD value) was measured at 412 nm, and the GSH concentration was calculated according to the formula in the instruction manual.(1)GSH (g/L) = [(measured OD value of the well-blank OD value of the well) × 307 × dilution factor]/13.6Note: 307 is the molecular weight of GSH and 13.6 is the mole fraction extinction coefficient of GSH.

### 2.11. mRNA High-Throughput Sequencing

U251 and U373 cells were each divided into two groups. The first group served as a control, and the second group overexpressed *PAX6*. Three replicates of each group were collected, and 1 × 10^7^ cells were collected for each sample, which were washed thoroughly with sterile PBS. Then, 1 mL of Trizol reagent was added and blown gently until the cells were completely lysed and then subsequently stored in dry ice. For the assay, RNA purity (OD260/280, OD260/230 ratio) and the RNA integrity index (RIN) were detected by NanoDrop™ One/OneC and Agilent 4200 TapeStation system. RNA was then precisely quantified by Life Invitrogen Qubit^®^ 3.0 Fluorescence Quantification Instrument. Samples with a RIN value of 6.0 or higher checked by the Agilent 4200 TapeStation System were regarded as qualified samples. After the samples passed the test, library construction was carried out. Since most of the mRNAs in eukaryotes have a polyA structure, magnetic beads with Oligo (DT) were used to capture mRNAs with a polyA structure, and then the first strand of cDNA was synthesized by using fragmented mRNAs as templates and random oligonucleotides as primers in a M-MuLV reverse transcriptase system. Then, the first strand of cDNA was degraded by RNaseH and DNA polymerase. The first strand of cDNA was synthesized in the M-MuLV reverse transcriptase system, the RNA was degraded by RNase H, and the second strand of cDNA was synthesized by dNTPs in the DNA polymerase I system. After purification and end repair, the cDNA was screened with AMPure XP beads at about 200 bp for PCR amplification, and the PCR products were purified using AMPure XP beads to obtain sequencing libraries, which were then merged into Illumina PE150 sequencing libraries according to the effective concentration and the target downstream data volume.

### 2.12. Linear Discriminant Analysis, Identification of Differentially Expressed Genes (DEGs), and Correlation Analysis

The glioma dataset (GSE151680 and GSE224727) was obtained from the GEO database (https://www.ncbi.nlm.nih.gov/geo/ accessed on 15 March 2025), and the microeco package from R (version 4.2.1) was used to perform linear discriminant analysis (LDA) of the effect sizes of each gene. The analysis strategy was set to one-to-many, the log LDA threshold for discriminant features was set to 3.0, and the *p*-value denoting significance was set to 0.05.

DEG screening was performed using R software (v4.2.1), specifically the DESeq2 package (threshold: corrected *p*-value < 0.05, |Log2FC| ≥ 2). Gene expression differences between the PAX6 overexpression group (PAX6-OE) and the control group (Con) were demonstrated by volcano plots and were screened for common DEGs in the U251 and U373 cell lines using the UpSetR package. Heatmaps were generated using the pheatmap package, demonstrating the expression of and using the common DEGs’ normalized Z-score. Genes were analyzed for KEGG pathway enrichment by the clusterProfiler package. Differential gene correlations were analyzed based on the igraph package calculation to obtain the Spearman correlation matrix. *p*-values were corrected using Benjamini–Hochberg’s test, with the threshold set to the Spearman correlation coefficient and corrected *p*-values of 0.6 and 0.05, respectively.

### 2.13. In Vivo Tumor Formation in Nude Mice

Eight 4-week-old nude mice were reared in a sterile environment and randomly divided into the PAX6-OE group and the Con group. Each mouse in the PAX6-OE group was injected subcutaneously with 5 × 10^6^ U251 cells overexpressing PAX6 (the concentration was adjusted with sterile PBS), and the Con group was injected with an equal amount of empty vector U251 cells. The body weight of the mice was monitored weekly, and the tumor tissues were peeled off after 4 weeks to measure their weight and volume. The protein expression levels of PAX6 and HIF-1α in the tumor tissues were detected by immunohistochemistry (IHC), and the expressions of PAX6, HIF-1α, and GPX4 were analyzed by Western blot. The animal experiments in this study have been approved by the Ethics Committee of the School of Basic Medical Science of Central South University, 2020KT-31.

Note: As a key transcription factor mediating cellular responses to hypoxic conditions, HIF-1α is typically upregulated in hypoxic environments. However, since the present study does not directly explore the association between HIF-1α and hypoxia, the subcutaneous tumor model in nude mice remains an appropriate choice.

### 2.14. Immunohistochemical Staining

Tumor tissues of nude mice from the Con group and PAX6-OE group were taken, fixed in 4% paraformaldehyde for 24 h, and then sequentially subjected to gradient ethanol dehydration, xylene clearing, and paraffin embedding; then, 4 μm thick continuous slices were prepared and adhered to antidetachment slides, and the slices were baked at 60 °C for 2 h. After the sections were deparaffinized by xylene and hydrated with gradient ethanol, 0.01 M citrate buffer (pH 6.0) was used for antigen retrieval, 3% H_2_O_2_ aqueous solution was used to block the endogenous peroxidase activity, and 5% normal goat serum was used to close the non-specific sites. The primary antibodies, PAX6 antibody and HIF-1α antibody, were titrated and incubated in a wet box at 4 °C overnight, followed by HRP-labeled secondary antibody incubated at 37 °C for 1 h. The color development time was controlled under a microscope with DAB color development solution (1–3 min), and water rinsing after hematoxylin restaining was performed until the section turned blue. The film was blocked with neutral gum after dehydration by gradient ethanol and clearing by xylene. The staining results were observed via light microscopy.

### 2.15. Statistical Methods

Data normality (Shapiro–Wilk test) and chi-squared (Bartlett’s test) tests were assessed by hypothesis testing, and *t*-tests (*p* > 0.05) or Mann–Whitney U tests (*p* ≤ 0.05) were chosen to analyze between-group differences according to the results. LDA, DEG analysis, KEGG analysis, and visualizing the statistical results were performed using the microeco package, DEseq2 package, and pheatmap package in R (v.2.1), and differences were considered statistically significant at *p* < 0.05.

## 3. Results

### 3.1. Decreased PAX6 Expression in Gliomas Exhibiting Aberrant Iron Metabolism

We assessed PAX6 expression of tumor tissues from 98 glioma patients across pathological grades 1–4 and 11 brain tissue samples (gray matter tissues of the brain containing glial cells) by tissue microarray technology. After excluding samples with heterogeneous staining, we retained 59 glioma tissue samples and 10 brain tissue samples from normal subjects. The density of brown particles served as the evaluation metric for the expression level. Immunohistochemistry showed that PAX6 expression progressively decreased with increasing glioma grade. Specifically, PAX6 expression in grade 3–4 gliomas was significantly lower than in normal tissues, while no statistically significant difference was observed between grade 1–2 tumors and normal samples (Figure 1A,B, * *p* < 0.05). In addition, qPCR and Western blot analysis further confirmed that *PAX6* mRNA and protein were significantly lower in glioma cell lines (U251 and U373) compared to normal human fetal glial cells (HMO6) (Figure 1C,D). To explore the effect size of genes in glioma cells, we performed LDA on a glioma dataset from the GEO database. The LDA results revealed a significantly increased expression of iron metabolism-related genes ferritin heavy chain 1 (*FTH1*), transferrin receptor (*TFRC*), and Acyl-CoA synthetase long-chain family member 4 (*ACSL4*), while GPX4 expression was decreased in the glioma samples compared to the controls (Figure 1E). This result suggested that iron metabolism-related gene expression was abnormal and that iron metabolism was altered in gliomas. The above results indicated that PAX6 downregulation in gliomas is accompanied by aberrant iron metabolism, suggesting that PAX6 may be involved in iron metabolism in glioma cells.

### 3.2. Overexpression of PAX6 Induces Ferroptosis in Glioma Cells

Ferroptosis is characterized by the accumulation of LPO in cell and organelle membranes under ROS. To preliminarily investigate whether PAX6 induces ferroptosis, we transiently transfected glioma cells (U251 and U373) with a PAX6 overexpression plasmid (pcDNA3.1-PAX6). LPO levels were significantly elevated in both cell lines compared to the control (empty vector-transfected) cells (Figure 2A,B), suggesting that PAX6 overexpression induces oxidative stress in glioma cells. To confirm ferroptosis involvement, we treated cells with Ferrostatin-1 (Fer-1), a specific ferroptosis inhibitor, which suppresses ferroptosis and may indirectly support tumor cell survival. U251 and U373 cells were divided into three groups, namely the control group transfected with an empty vector (pcDNA3.1-Con), the PAX6 overexpression group, and the intervention group with PAX6 overexpression plus 60 nM Fer-1. Cell proliferation assays revealed that PAX6 overexpression markedly inhibited cell growth, and this effect was partially reversed by Fer-1 (Figure 2C,D). Furthermore, overexpression of PAX6 can increase the concentration of Fe^2+^ within glioma cells, while overexpression of HIF-1α has the opposite effect (Figure 2E,F). These results indicate that PAX6 suppresses glioma cell proliferation, at least in part, by inducing ferroptosis.

### 3.3. PAX6 Overexpression Alters Expression of Oxidative Stress and Iron Metabolism Genes in Glioma Cells

High-throughput RNA sequencing (RNA-seq) was performed to profile gene expression changes following PAX6 overexpression in U251 and U373 cells. Volcano plots revealed 187 differentially expressed genes (DEGs) in U251 cells (110 upregulated, 77 downregulated) and 213 DEGs in U373 cells (149 upregulated, 6 downregulated) compared to the controls (Figure 3A,B). Cross-analysis identified 142 common DEGs in both cell lines (Figure 3C), suggesting that these 142 differential genes may play an important role in gliomas in PAX6-OE. Heatmap analysis of these DEGs showed significant changes in key regulators such as Ferritin Light Chain (*FTL*), Hypoxia-Inducible Factor 3 Alpha (*HIF-3α*), *TFRC, HIF-1αN* and *GPX4* (Figure 3D). KEGG pathway enrichment of these DEGs indicated involvement in oxidative phosphorylation and glutathione metabolism—pathways critical for redox balance and iron metabolism (Figure 3E). Further analysis revealed that *PAX6* overexpression upregulated ROS-related genes including glutamate–cysteine ligase catalytic (*GCLC*), superoxide dismutase 2 (*SOD2*), and thioredoxin reductase 1 (*TXNRD1*), reinforcing its role in promoting oxidative stress (Figure 3F). The aforementioned data strongly indicate that *PAX6* overexpression may induce redox imbalance in glioma cells by disrupting iron metabolism equilibrium and boosting ROS generation.

### 3.4. PAX6 Suppresses HIF-1α Expression via ROS Generation in Glioma Cells

To determine the relationship between PAX6, ROS, and HIF-1α, we used the ROS-sensitive probe DCFH-DA. Fluorescence microscopy revealed that ROS levels were significantly increased in U251 and U373 cells overexpressing PAX6 (Figure 4A). Given that HIF-1α is recognized for its inhibitory effect on ferroptosis across many malignancies, we explored whether the PAX6-mediated increase in ROS was related to HIF-1α. We next examined HIF-1α expression in human glioma tissue arrays. Immunohistochemistry demonstrated that HIF-1α expression increased with glioma grade. While grade 1 tumors showed no significant difference compared to normal tissue, grades 2–4 exhibited markedly elevated HIF-1α levels (Figure 4B,C). We conducted a correlation analysis of the expression of PAX6 and HIF-1α in each tumor sample and found that there was a negative correlation between the two (Figure 4D). Furthermore, overexpression of PAX6 can significantly inhibit the expression of HIF-1α in U251 and U373 cells (Figure 4E). Consistent with these findings, both qPCR and Western blot confirmed significantly higher HIF-1α expression in U251 and U373 cells relative to HMO6 cells (Figure 4F,G). Together, these results suggest that PAX6 downregulates HIF-1α by enhancing ROS production, which may be a key mechanism through which it promotes ferroptosis.

### 3.5. PAX6 Induces Ferroptosis in Glioma Cells by Inhibiting HIF-1α

To further dissect the regulatory relationships among PAX6, ROS, iron metabolism, and ferroptosis, we conducted Spearman correlation analysis. The results showed strong correlations among ROS- and iron metabolism-related genes. Notably, PAX6 and HIF-1αn (the inhibitor of HIF-1α) expression were negatively correlated with GPX4 (Figure 5A). To validate these relationships, U251 and U373 cells were divided into four groups, namely the control, HIF-1α overexpression group, PAX6 overexpression, and co-overexpression of both PAX6 and HIF-1α. It was found that overexpression of HIF-1α enhanced glioma cell proliferation and partially rescued the growth inhibition induced by PAX6 (Figure 5B,C). Additionally, further investigation of the regulatory effects of HIF-1α on ferroptosis-related molecules in glioma cells revealed that HIF-1α overexpression decreased LPO levels and increased GSH concentrations, counteracting the ferroptosis-promoting effects of PAX6 (Figure 5D–G). Since GPX4 is a key molecule in regulating ferroptosis, we examined the protein expression level of GPX4 in the above groups of cells by Western blot, and the results showed that PAX6 overexpression significantly suppressed GPX4 protein expression, while HIF-1α co-expression reversed this suppression to some extent, and overexpression of HIF-1α alone can promote the expression of GPX4. (Figure 5H,I). These results collectively demonstrate that PAX6 promotes ferroptosis in glioma cells primarily by downregulating HIF-1α.

### 3.6. In Vivo Validation: PAX6 Overexpression Inhibits Glioma Growth by Promoting Ferroptosis

To further investigate the function and mechanism of the PAX6 in vivo, stable U251-PAX6-OE cells and U251-Con cells were injected into the subcutaneous tissue of nude mice. Consequently, the growth rate of PAX6-overexpressing tumors was partially reduced, as evidenced by changes in tumor weights and volumes (Figure 6A–D), demonstrating the inhibitory effect of PAX6 overexpression on glioma growth in vivo. Immunohistochemical staining analysis of the excised tumor tissues revealed that PAX6 overexpression also markedly suppressed the expression of HIF-1α in glioma cells in vivo, consistent with the findings from in vitro experiments (Figure 6E). Western blot results on tumor tissues demonstrated that PAX6 overexpression can reduce the expression of the GPX4 protein in glioma cells (Figure 6F), indicating the potential role of PAX6 in inducing ferroptosis in glioma cells. Thus, our findings validate that activating PAX6 can impede glioma cell growth both in vitro and in vivo, with one mechanism inducing ferroptosis in glioma cells, suggesting that PAX6 could serve as a promising target for glioma.

## 4. Discussion

Gliomas are the most common primary malignant tumors of the central nervous system, characterized by aggressive growth and resistance to conventional therapies. These biological features present major challenges to effective treatment and represent a significant public health burden. At the molecular level, glioma development involves genomic instability driven by the activation of oncogenes and the inactivation of tumor suppressor genes—a hallmark of many solid tumors. Previous studies have suggested that PAX6 also functions a tumor suppressor in gliomas [34,35]. Our earlier work confirmed that PAX6 expression is reduced in gliomas tissues and that its overexpression inhibits glioma cell proliferation and invasion [17]; however, the molecular mechanism by which PAX6 regulates malignant phenotypes in the glioma microenvironment has not been fully elucidated, as PAX6 exhibits functional heterogeneity across different tumor types. For instance, in epithelial-derived tumors such as non-small-cell lung cancer, PAX6 is overexpressed and promotes tumor proliferation [36,37]. In contrast, in gliomas and prostate cancer, PAX6 functions as a tumor suppressor [38,39]. These divergent roles highlight the importance of context-specific regulatory mechanisms, including signaling pathway differences. Understanding how PAX6 operates specifically in gliomas is therefore essential for elucidating the molecular pathological basis of glioma development and developing targeted therapeutic strategies.

In this study, we found that PAX6 expression decreased with increasing glioma pathologic grade, supporting its role as a tumor suppressor. Bioinformatics analysis using LDA of glioma datasets from the GEO database revealed widespread dysregulation of iron metabolism-related genes, suggesting a possible regulatory relationship between low PAX6 expression and iron metabolism disruption.

While no prior reports directly demonstrate that PAX6 regulates ferroptosis in tumor cells, studies have been reported to indirectly support the involvement of PAX6 in redox regulation. Chang et al. reported that PAX6 overexpression significantly increased ROS expression in glioblastoma [40], and Hegge et al. found that PAX6 knockdown enhanced glioma cell resistance to oxidative stress induced by H_2_O_2_ [41]. These studies indicate that PAX6 can promote ROS generation in glioma cells and influence redox balance. Our transcriptomic analysis further supported this hypothesis. Overexpression of PAX6 upregulated ROS-related genes and disrupted pathways involved in oxidative phosphorylation and glutathione metabolism. These alterations contributed to ROS accumulation and coincided with the downregulation of HIF-1α. Biostatistics analysis also revealed a negative correlation between PAX6 and several members of the HIF family (HIF-1αN and HIF-3α). Furthermore, PAX6 overexpression influenced various genes associated with ROS generation and initiated a cascade of altered oxidative stress. This finding demonstrated that PAX6 could regulate HIF-1α expression via ROS, thereby inducing ferroptosis in glioma cells. Moreover, we discovered that PAX6 expression showed negative correlations with GPX4, a key ferroptosis-regulating enzyme, while enhancing lipid peroxides (LPOs) and inhibiting glutathione (GSH) synthesis, which demonstrated that PAX6 could inhibit the malignant progression of glioma cells by activating the ferroptosis pathway at the molecular level.

Solid tumors like gliomas often reside in a chronically hypoxic microenvironment. To sustain cellular viability, tumor cells maintain energy metabolism through the Warburg effect [42], in which HIF-1α, as a principal regulator, enhances lactate production by activating glucose transporter (GLUT1) and hexokinase 2 (HK2). Additionally, HIF-1α can also promote tumor proliferation and invasion by modulating the expression of the proto-oncogene c-myc and the matrix metalloproteinase MMP9 [43,44,45,46]. Recent studies have found that HIF-1α can promote tumor cell survival and proliferation by regulating key points of iron metabolism to promote tumor cell resistance to ferroptosis. Zheng et al. demonstrated that HIF-1α overexpression in non-small-cell lung cancer elevated GPX4 expression, while HIF-1α silencing induced ferroptosis and inhibited tumor proliferation in lung cancer cells [32]; Gao et al. reported that elevated HIF-1α in hepatocellular carcinoma cells significantly inhibited sorafenib-induced cellular ferroptosis [47]. The association between HIF-1α and the regulation of ferroptosis in gliomas has not been reported. Our study aligns with these findings, showing that PAX6 overexpression suppressed the expression of the *HIF-1α* gene and protein in glioma cells, whereas concurrent overexpression of PAX6 and HIF-1α mitigated the ferroptosis impact generated by PAX6.

This study proposes that PAX6 induces (ROS) accumulation in glioma cells, which subsequently suppresses HIF-1α expression and promotes ferroptosis. However, limitations persist, as we have not completely clarified the mechanisms by which PAX6 regulates HIF-1α nor have we fully explored the specific interactions between oxidative stress and iron metabolism in gliomas, which requires further investigation. Notably, the correlation between PAX6 and HIF-1α has not been previously reported, and our findings suggest a potential negative regulatory relationship between these two factors in gliomas. Elucidating the underlying mechanisms of this interaction will be a key focus of future research. Moreover, the expression level of PAX6 varies significantly among patients with different grades of gliomas (mainly grades 2–4, classified according to WHO CNS5 Tumour Classification (2021)). If further research is to be conducted on the potential therapeutic targets of PAX6 in gliomas, it is a very necessary step to verify its efficacy in clinical glioma patients.

In summary, this study illustrates that PAX6 promotes ferroptosis in glioma cells by suppressing HIF-1α by regulating intracellular ROS and oxidative stress. These findings not only reveal the important biological functions of PAX6 in glioma development and progression but also highlight the critical role of PAX6 in regulating HIF-1α and ferroptosis during the pathological process of glioma. This provides a theoretical basis for exploring PAX6-targeted, ferroptosis-inducing gene therapy strategies.

## 5. Conclusions

This study reveals a novel tumor-suppressive mechanism of PAX6 in gliomas, whereby it inhibits HIF-1α to induce ferroptosis, ultimately suppressing tumor growth. Our findings enhance the understanding of glioma pathogenesis and the role of PAX6, while also providing an experimental foundation for further investigating the PAX6/HIF-1α axis. Nevertheless, given the complexity of PAX6 signaling, many underlying mechanisms remain unclear. Thus, further research is warranted to explore PAX6 as a potential therapeutic target for glioma.

## Figures and Tables

**Figure 1 biomolecules-15-01462-f001:**
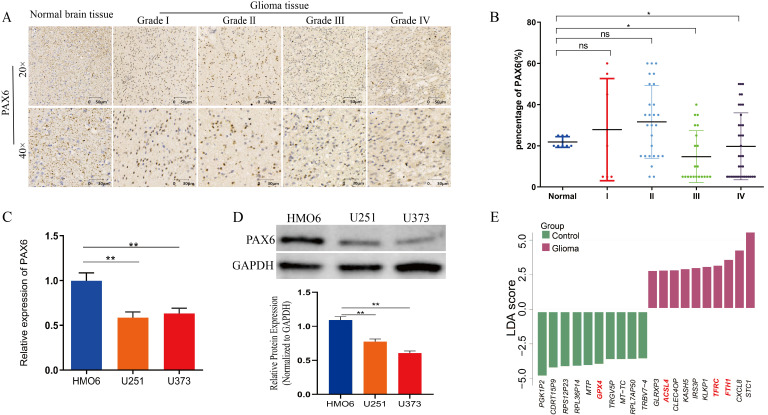
Reduced PAX6 expression and abnormal iron metabolism are present in gliomas. (**A**) PAX6 expression in normal brain tissues and tumor tissues of glioma patients, which are detected by immunohistochemistry; the number of brown particles represents the protein expression level. (**B**) Comparison of PAX6 expression levels in normal brain tissues and tumor tissues of glioma patients with different grades of gliomas. (**C**) Detection of mRNA expression levels of *PAX6* in glioma cells (U251/U373) and normal glial cells (HMO6) using qPCR. (**D**) Detection of protein expression levels of PAX6 in glioma cells (U251/U373) and normal glial cells (HMO6) using Western blot. (**E**) Linear discriminant analysis of gene expression in glioma patients showing abnormal expression of iron metabolism-related genes. Differences between groups were analyzed using unpaired *t*-test, and both the qPCR and Western blot experiments were repeated three times with data is presented as X ± SD (ns *p* > 0.05, * *p* < 0.05, ** *p* < 0.01). The original images are in the Appendix A.

**Figure 2 biomolecules-15-01462-f002:**
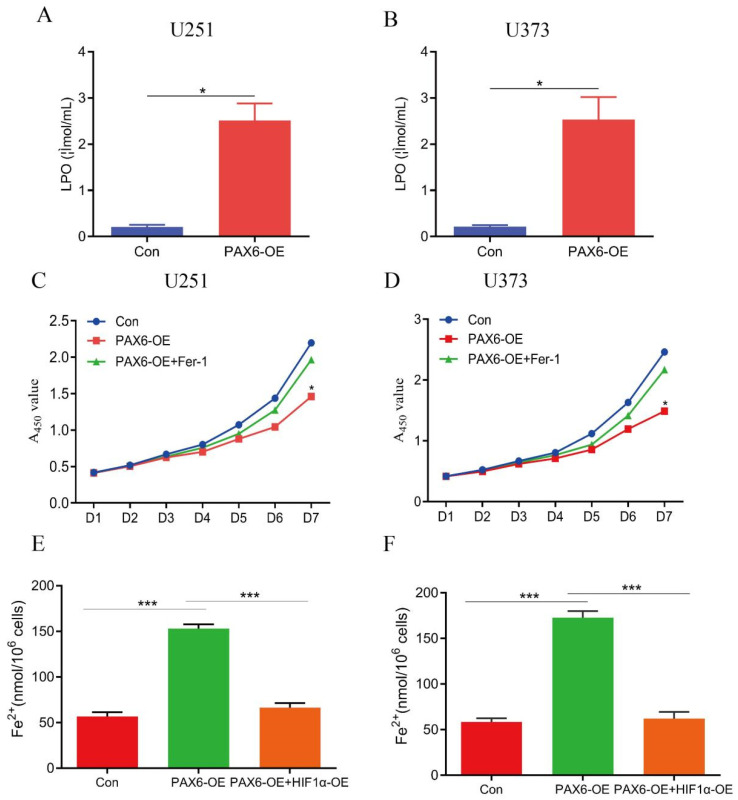
PAX6 overexpression induces ferroptosis in glioma cells. (**A**,**B**) LPO expression levels in PAX6 overexpressing glioma cells. (**C**,**D**) PAX6 overexpression inhibited proliferative ability of glioma cells, and this inhibitory effect could be blocked by ferroptosis inhibitor Fer-1. Con: control group; PAX6-OE: PAX6 overexpression group; PAX6-OE + Fer-1: PAX6 overexpression combined with ferroptosis inhibitor Fer-1 treatment group. (**E**,**F**) Quantification of Fe^2+^ concentration (mmol per 10^6^ cells) in cells with PAX6 overexpression (PAX6-OE) and in cells with combined PAX6 and HIF1α overexpression (PAX6-OE + HIF1α-OE). (* *p* < 0.05, *** *p* < 0.001).

**Figure 3 biomolecules-15-01462-f003:**
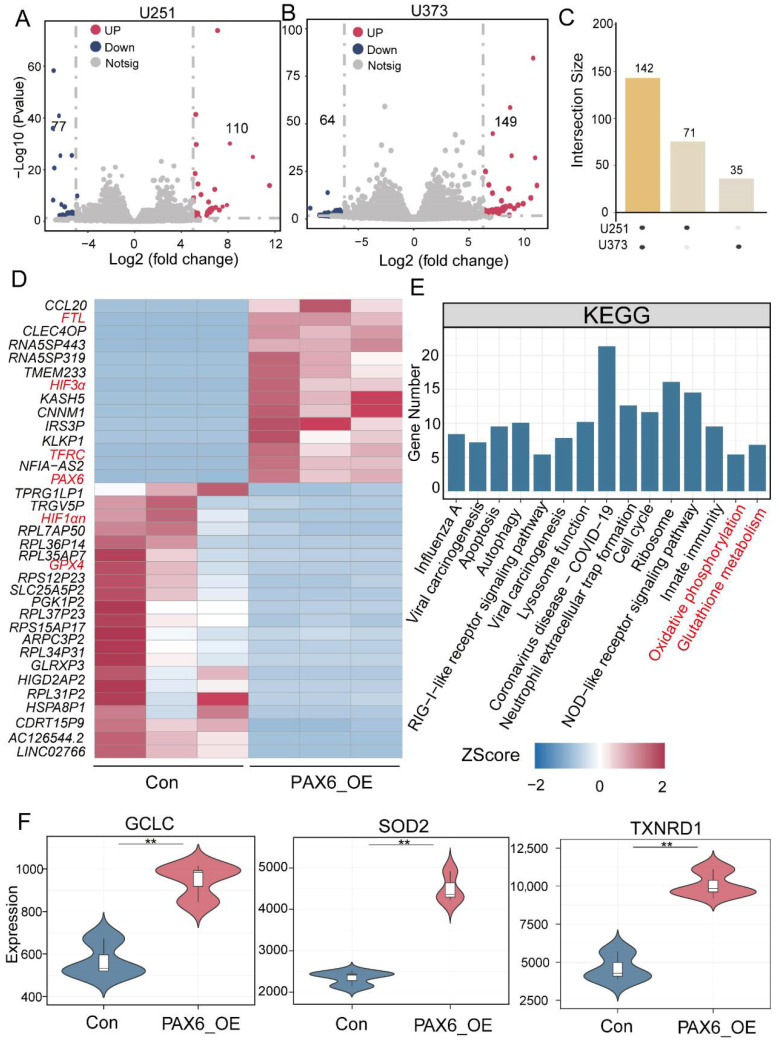
*PAX6* overexpression in glioma cells affects the expression of genes related to oxidative stress and iron metabolism. (**A**,**B**) Differential genes of U251 and U373 in *PAX6* overexpressed glioma cells compared to the control. The color of the volcano plot nodes indicates the type of their changes; blue indicates downregulation, red indicates upregulation, and gray indicates no significant difference between the groups. (**C**) Upset plots screened for 142 common differential genes in two types of glioma cells, U251 and U373l. (**D**) Heatmap demonstrating the gene expression of the 142 common differential genes in the Con group versus the PAX6_OE group. (**E**) The expression of the 142 common differential genes overexpressed by *PAX6* were enriched to a total of 15 pathways by KEGG enrichment analysis. (**F**) The expression of oxidative stress-related genes *GCLC*, *SOD2*, and *TXNRD1* in the PAX6 overexpression group compared to the Con group. The figures (**D**–**F**) are the analysis results for the glioma cell lines. Con: control group; PAX6-OE: PAX6 overexpression glioma cell group (** *p* < 0.01).

**Figure 4 biomolecules-15-01462-f004:**
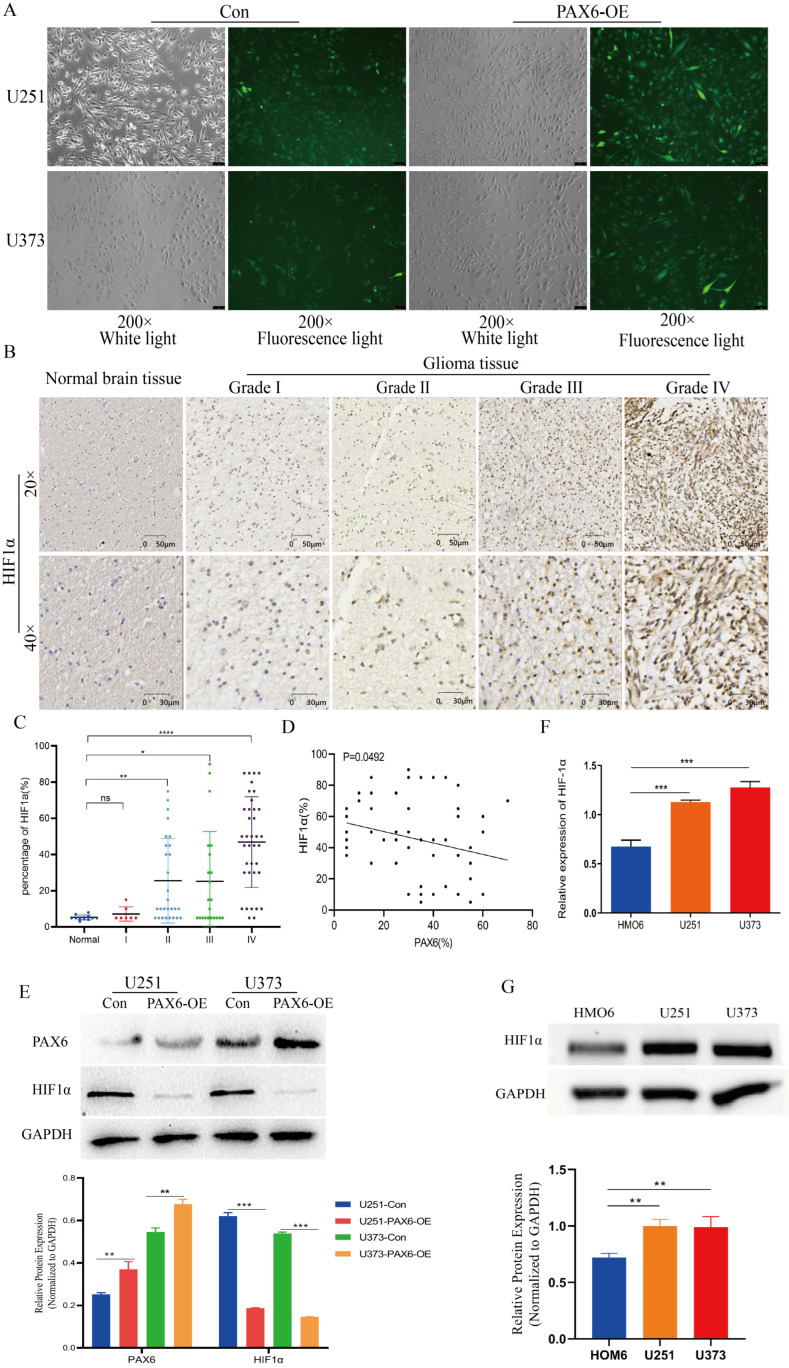
PAX6 overexpression inhibits HIF-1α expression in glioma cells. (**A**) ROS fluorescent probe DCFH-DA detects ROS expression levels in normal glioma cells versus overexpressed PAX6 glioma cells. (**B**) HIF-1α expression in normal brain tissues and tumor tissues of glioma patients with the number of brown particles representing the protein expression level. (**C**) Comparison of HIF-1α expression in normal brain tissues and tumor tissues of glioma patients of different grades. (**D**) Correlation analysis between PAX6 and HIF-1α expression in glioma tissues (Pearson correlation test, *p* = 0.0492). (**E**) Western blot analysis of PAX6 and HIF-1α expression in U251 and U373 cells with PAX6 overexpression (PAX6-OE). (**F**) Detection of protein expression level of HIF-1α in HMO6, U251, and U373 cells. (**G**) Detection of mRNA expression level of HIF-1α in HMO6, U251, and U373 cells. Student’s *t*-test was used to analyze the statistical differences between any two groups of measurement data. (ns *p* > 0.05, * *p* < 0.05, ** *p* < 0.01, *** *p* < 0.001, **** *p* < 0.0001). The original images are in the Appendix A.

**Figure 5 biomolecules-15-01462-f005:**
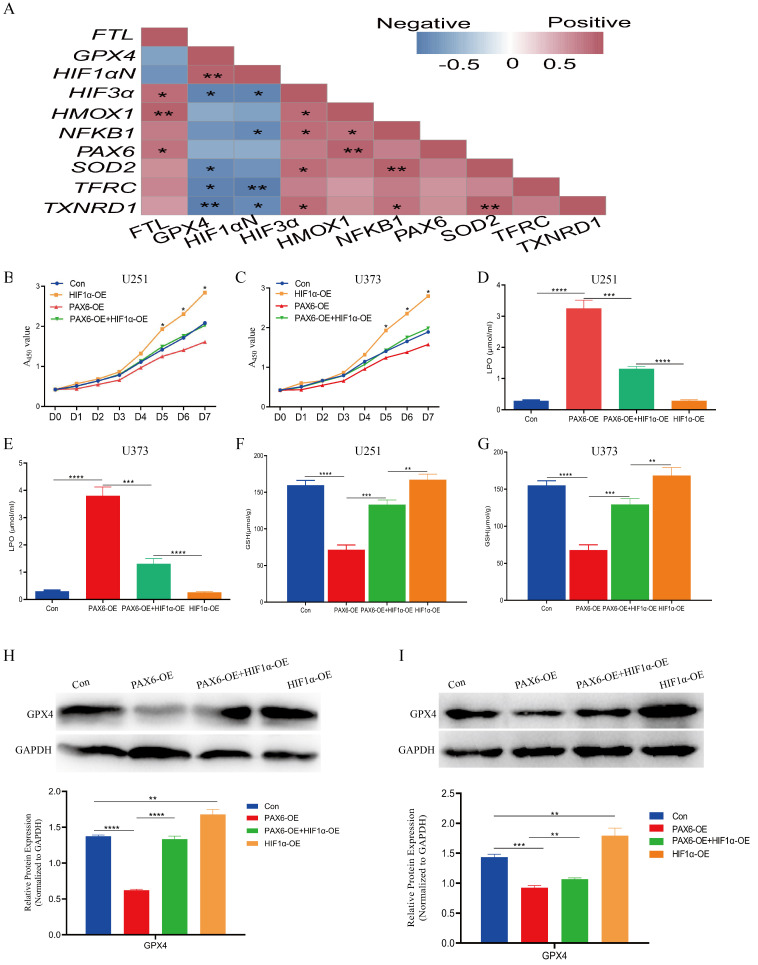
Overexpression of PAX6 promotes ferroptosis in glioma cells by inhibiting high HIF-1α expression. (**A**) Spearman correlation analysis of the correlation between oxidative stress genes and iron metabolism genes at PAX6-OE. (**B**,**C**) CCK-8 assay to detect the proliferative ability of glioma cells U251 and U373 cells under different treatment conditions. (**D**,**E**) LPO expression level of U251 and U373 cells detected under different treatment conditions. (**F**,**G**) ELISA method to detect the expression level of GSH in U251 and U373 cells under different treatment conditions. (**H**,**I**) Western blot to detect the level of GPX4 in U251 and U373 cells under different treatment conditions. Student’s *t*-test was used to analyze the statistical differences between any two groups of measurement data. Con: control group; PAX6-OE: PAX6 overexpression group; PAX6-OE + HIF-1α-OE: PAX6 and HIF-1α co-overexpression group; HIF-1α-OE: HIF-1α overexpression group (* *p* < 0.05, ** *p* < 0.01, *** *p* < 0.001, **** *p* < 0.0001). The original images are in the Appendix A.

**Figure 6 biomolecules-15-01462-f006:**
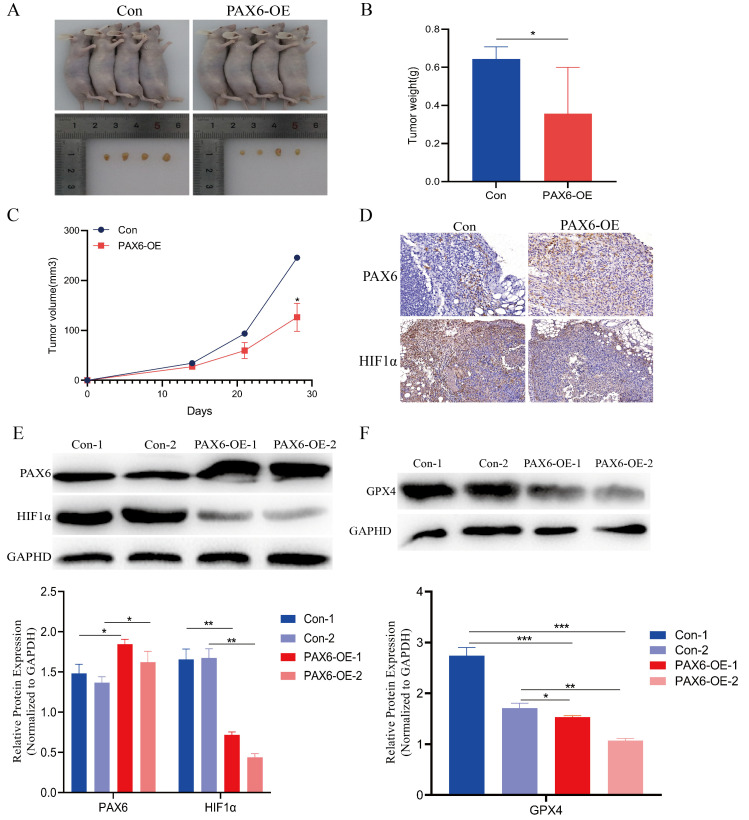
The validation of the mechanism of action of PAX6 in inhibiting HIF-1α to promote ferroptosis in glioma cells. (**A**) Tumorigenesis was induced by implanting U251-PAX6 and U251-Con cells into nude mice, respectively. (**B**) Comparison of tumor weight after tumorigenesis of U251-PAX6 and U251-Con cells. (**C**) Comparison of tumor volume after tumorigenesis of U251-PAX6 and U251-Con cells. (**D**) Immunohistochemistry was performed to detect the expression of HIF-1α in the tumor tissues of nude mice in the Con group and the PAX6-OE group. PAX6 and HIF-1α expression. (**E**) Western blot analysis of PAX6 and HIF-1α expression in the tumor samples of two nude mice from the Con group and PAX6-OE group. (**F**) Western blot detection of GPX4 expression in the tumor samples of two nude mice from the Con group and PAX6-OE group (* *p* < 0.05, ** *p* < 0.01, *** *p* < 0.001). The original images are in the Appendix A.

**Table 1 biomolecules-15-01462-t001:** Primer sequences for *PAX6* and *HIF-1α*.

Primer Name	Primer Sequences (5′-3′)
*PAX6*-F	CCGTGTGCCTCAACCGTA
*PAX6*-R	CACGGTTTACTGGGTCTGG
*HIF-1α*-F	TCGGCGAAGTAAAGAATC
*HIF-1α*-R	TTCCTCACACGCAAATAG
*GAPDH*-F	CACCAGGGCTGCTTTTAACTC
*GAPDH*-R	GAAGATGGTGATGGGATTTC

Note: F: the forword primer; R: the revised primer.

## Data Availability

The original contributions presented in this study are included in the article/Appendix A. Further inquiries can be directed to the corresponding authors (Q.Z. or B.H.).

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
