# Peer review of "PAX6 Downregulation Triggers HIF-1α-Mediated Ferroptosis in Glioma Cells"

_biomolecules, 2025, doi:10.3390/biom15101462_

Round 1

Reviewer 1 Report (Previous Reviewer 1)

Comments and Suggestions for Authors

The manuscript "PAX6 Downregulation Triggers HIF-1α-Mediated Ferroptosis 2 
in Glioma Cells " Luo Q et al. describes the PAX6 dysregulation in modulating HIF-1α and 110 
ferroptosis in gliomas. Though the previous comments has been addressed by the authors, some issues still needs to be looked after.

In the results section,

Figure 1B: There are outliers (2) in the control samples that are included in analysis of the data giving a higher median which is resulting in significant difference of the expression pattern with glioma samples. The outliers should be excluded or more control samples should be included and the analysis should be performed again.

Figure 4A/B: the magnification (200X and 400X) need to be checked, is it 20X and 40X?

Figure 4G: The significance of the plotted data of western blot does not look like p<0.001 between  HMO6 cells with U251 and U373.

Similarly in Figure 5H/I, difference between con and PAX6OE-HIF1a-OE does not look like p<0.001. Please clarify the statistical tests and values

Author Response

Response to the Reviewer 1

Dear Reviewer,

Thank you very much for your meticulous review and guidance. We have made comprehensive revisions to our manuscript based on your comments. Now, we are providing point-by-point responses. Please have review them.

Q1 Figure 1B: There are outliers (2) in the control samples that are included in analysis of the data giving a higher median which is resulting in significant difference of the expression pattern with glioma samples. The outliers should be excluded or more control samples should be included and the analysis should be performed again.

A1 Thank you very much for your suggestion. We carefully checked the data and found that there were indeed two samples in the control group whose values were significantly abnormal. However, after careful analysis, we believe that these two samples still fall within the normal range. This is because the expression of PAX6 in normal brain tissue is already higher than that in glioma tissue. In fact, the other values in the control group should also be higher. However, due to the high heterogeneity of PAX6 in different populations and the difficulty in obtaining normal brain tissue samples (the normal control tissues in the tissue chips we purchased might be cancerous adjacent tissues from glioma patients), it is indeed difficult to increase samples for statistical analysis. Additionally, there have been many studies reporting low expression levels of PAX6 in glioma (For example, Antagonistic modulation of gliomagenesis by Pax6 and Olig2 in PDGF-induced oligodendroglioma, PMID: 22514120), and we have also mentioned this in our manuscipt. Given this, considering that our control group samples are already very few, after careful consideration, we hope to retain these two values. Thanks again.

Q2 Figure 4A/B: the magnification (200X and 400X) need to be checked, is it 20X and 40X?

A2 Thank you for your careful review. We rechecked the magnification of these two figures. First, Fig 4A shows the detection of ROS in cells by using a fluorescence microscope. We used a 20X object glass for the detection, and the microscope itself has a 10X eyepiece, so the 200X is correct. Secondly, Fig 4B is an immunohistochemical result of glioma tissue chips. This map was also photographed using a low-power lens (20X) and a high-power lens (40X) during scanning, but there is also a 10X eyepiece, so 200X and 400X are correct.

Q3 Figure 4G: The significance of the plotted data of western blot does not look like p<0.001 between  HMO6 cells with U251 and U373.

Similarly in Figure 5H/I, difference between con and PAX6OE-HIF1a-OE does not look like p<0.001. Please clarify the statistical tests and values.

A3 Thank you very much for your guidance. We have verified the data of these figures. Firstly, Fig 4G is correct and indeed p < 0.001. Secondly, for Fig 5H and 5I, we repeated the experiment, updated the more optimized western blot band structure, and explained the testing method in the legends of Fig 4 and Fig 5. Here are the statistical analysis values related to Fig 4G, Fig 5H, and Fig 5I, including t values and P values. We hope that the above data analysis results can meet the requirements of Biomolecules. Thank you.

Fig 4G

t

P

HMO6 vs U251

12.090

0.0003

HMO6 vs U251

8.666

0.0010

Fig 5H

t

P

Con vs PAX6-OE

65.700

<0.0001

Con vs HIF1α-OE

7.549

0.0016

PAX6-OE vs PAX6-OE+HIF1α-OE

28.170

<0.0001

Fig 5I

t

P

Con vs PAX6-OE

17.050

<0.0001

Con vs HIF1α-OE

90.260

<0.0001

PAX6-OE vs PAX6-OE+HIF1α-OE

31.420

<0.0001

Thank you and best regards.

Yours sincerely

Quan Zhu

Reviewer 2 Report (Previous Reviewer 2)

Comments and Suggestions for Authors

The authors have provided a comprehensive response to this reviewer's initial review and furthermore, their explanations are acceptable overall. As a result, and also in response to other reviews, the manuscript is significantly improved.

However, a few additional revisions are necessary to frame either the experimental design and/or study conclusions in their most appropriate context for the readers, who will be primarily from the biomedical field of neuro-oncology.

  1. The revised Abstract should qualify the last sentence of the Conclusions (line 36) by adding a word such as "downstream of" prior to "PAX6". The authors have nicely phrased this on the last sentence of the Discussion (lines 556-557) since the future goal is not to directly target the transcription factor PAX6 as a treatment strategy.
  2. The revised Methods should explicitly state that this study is not directly studying the relationship between HIF1α and hypoxia; hence, the use of subcutaneous in vivo models is suitable.
  3. The revised Methods should specifically state (line 119) whether the anti-PAX6 antibody from Cusabio was polyclonal or monoclonal. Similarly (line 141) for the anti-PAX6 antibody from Millipore. These details will permit other groups to reproduce these studies independently. Note to the authors: Rest assured, this reviewer has no intention of doing so!
  4. The RIN cut-off presented in the revised manuscript is actually the 28S/18S ratio of 1.0 (line 241). Please replace with a RIN >6.0 as you have stated in your response to the reviewer.
  5. In a "future directions" section of the revised Discussion, the authors should suggest that corroboration in clinical samples of gliomas (grades 2-4) assessed under the WHO CNS5 Tumour Classification (2021) is an important step.

Author Response

Response to the Reviewer 2

Dear Reviewer,

Thank you very much for your meticulous review and guidance. We have made comprehensive revisions to our manuscript based on your comments. Now, we are providing point-by-point responses. Please have review them.

Q1 The revised Abstract should qualify the last sentence of the Conclusions (line 36) by adding a word such as "downstream of" prior to "PAX6". The authors have nicely phrased this on the last sentence of the Discussion (lines 556-557) since the future goal is not to directly target the transcription factor PAX6 as a treatment strategy.

A1 Thank you for your guidance. We have rewritten the conclusion of the abstract according to your suggestions.

Conclusions: Downstream of PAX6 plays an important role in regulating ferroptosis in glioma cells. Our research provides a reference basis for a deeper understanding of the role of PAX6 in ferroptosis of glioma (See lines 36-38).

Q2 The revised Methods should explicitly state that this study is not directly studying the relationship between HIF1α and hypoxia; hence, the use of subcutaneous in vivo models is suitable.

A2 Thank you very much for your guidance. We have added the corresponding content in the method section.

Note: As a key transcription factor mediating cellular responses to hypoxic conditions, HIF-1α is typically upregulated in hypoxic environments. However, since the present study does not directly explore the association between HIF-1α and hypoxia, the subcutaneous tumor model in nude mice remains an appropriate choice (See lines 294-297).

Q3 The revised Methods should specifically state (line 119) whether the anti-PAX6 antibody from Cusabio was polyclonal or monoclonal. Similarly (line 141) for the anti-PAX6 antibody from Millipore. These details will permit other groups to reproduce these studies independently. Note to the authors: Rest assured, this reviewer has no intention of doing so!

A3 Thank you very much for your guidance. We have added the corresponding content in the method section.

All the antibodies mentioned here are rabbit-derived polyclonal antibodies (See lines 124-125).

These two antibodies are both rabbit-derived monoclonal antibodies (See line 146).

Q4 The RIN cut-off presented in the revised manuscript is actually the 28S/18S ratio of 1.0 (line 241). Please replace with a RIN >6.0 as you have stated in your response to the reviewer.

A4 Thank you very much for your guidance. We have added the corresponding content in the method section.

Samples with a RIN value of 6.0 or higher checked by the Agilent 4200 TapeStation System were regarded as qualified samples (See line 249).

Q5 In a "future directions" section of the revised Discussion, the authors should suggest that corroboration in clinical samples of gliomas (grades 2-4) assessed under the WHO CNS5 Tumour Classification (2021) is an important step.

A5 Thank you very much for your guidance. We have added the corresponding content in the method section.

Moreover, the expression level of PAX6 varies significantly among patients with different grades of gliomas (mainly grades 2-4, classified according to WHO CNS5 Tumour Classification (2021)). If further research is to be conducted on the potential therapeutic targets of PAX6 in gliomas, it is a very necessary step to verify its efficacy in clinical glioma patients (See lines 572-576).

Thank you and best regards.

Yours sincerely

Quan Zhu

Round 2

Reviewer 1 Report (Previous Reviewer 1)

Comments and Suggestions for Authors

The authors Luo Q et al. of the manuscript "PAX6 Downregulation Triggers HIF-1α-Mediated Ferroptosis in Glioma Cells" have revised the manuscript substantially however, a few clarifications still need to be provided by the authors:-

Figure 1b, the control samples seems to have two outliers that are included in the analysis because of which the mean/ median seems to be higher than the actual value giving the significant results. The authors can consider reanalysing the data after removing the two controls or enroll more controls for the analysis.

Figure 4G, Similarly the significance of the plotted graph does not seem to be p<0.001. Kindly check again.

Figure 5H/I: the significance of plotted western blot graphs are not convincing, please mention the tests and samples for which the  comparison has been done.

Figure 4, Please check the magnification, it seems to be 20X and 40X instead of 200and 400.

Author Response

Thank you very much for your meticulous review and guidance. We have made comprehensive revisions to our manuscript based on your comments. Now, we are providing point-by-point responses. Please have review them.

Q1 Figure 1b, the control samples seems to have two outliers that are included in the analysis because of which the mean/ median seems to be higher than the actual value giving the significant results. The authors can consider reanalysing the data after removing the two controls or enroll more controls for the analysis.

A1 Thank you very much for your suggestion. We reanalyzed the data and removed two outliers from the control group as well as one outlier from Group III (PAX6% = 70%). We considered that grade III gliomas are high-grade gliomas and these patients generally have low expression of PAX6, so we removed this outlier from Group III. Subsequently, through statistical analysis, we found that the conclusion remained unchanged. However, if we did not remove this outlier from Group III, only Group IV and the control group would have statistical differences. Additionally, since we purchased glioma tissue microarrays, there were only a small number of normal brain glial tissues (possibly from adjacent tissues of tumors), and it is very difficult to obtain a large amount of brain tissue, so it is hard for us to expand the sample size for testing. Thank you again for your review.

Q2 Figure 4G, Similarly the significance of the plotted graph does not seem to be p<0.001. Kindly check again.

A2 Thank you for your correct guidance. We re-checked and re-scanned the grayscale values of the experimental results in Fig 4G three times and concluded that the group of HMO6 vs U251 and HMO6 vs U373 were both p < 0.01 (**) rather than p < 0.001 (***). We sincerely request that you review it again.

Q3 Figure 5H/I: the significance of plotted western blot graphs are not convincing, please mention the tests and samples for which the comparison has been done.

A3 Thank you very much for your careful review. We have re-examined these two figures and re-conducted the experiment for Fig 5I, replacing it with an optimized Western blot image. We have used an independent sample t-test (Student’s t test) for the gray values of different groups, which has been marked in the manuscript (See the legend of Fig 5). Please review it again. Thank you for your hard work.

Q4 Figure 4, Please check the magnification, it seems to be 20X and 40X instead of 200 and 400.

A4 Thank you very much for your suggestion. We have once again thoroughly verified the magnification factor. For Fig 4A, we used Leica microscope for the test, and the final magnification factor is the product of the eyepiece and objective magnification. Therefore, our magnification factor is correct and indeed 200 times. However, for the immunohistochemical figure (Fig 4B), your suggestion was very correct. We used the scanning software for the test, and it was actually magnified by 20 times and 40 times, not 200 times and 400 times. We also corrected the magnification factor of Fig 1A at the same time. We sincerely apologize for this oversight and once again thank you for your careful review and consideration for us.

Thank you and best regards.

Yours sincerely

Quan Zhu

This manuscript is a resubmission of an earlier submission. The following is a list of the peer review reports and author responses from that submission.

Round 1

Reviewer 1 Report

Comments and Suggestions for Authors
  • There is need of revision of English in various sentences so as to improve the understanding of manuscript such as in line 180 closed should be replaced with blocked, line 189 inoculated with seeded, line 219 gently blown etc.
  • In “Establishment of Stable Cell Lines” sections, authors should specify what is meant by logarithmic growth phase of cell.
  • Authors are suggested to add the sources of plasmids.
  • Authors are suggested to mention the no of replicates for assays such as RT-PCR, CCK-8 assays, Lipid Peroxides (LPO) Detection, etc.
  • The details of the Lipid Peroxides (LPO) Detection, Glutathione (GSH) assay kit should be mentioned along with the formula used for calculation.
  • In statistical analysis “Data normality ….. according to the results” should be more elaborated.
  • In results section 3.1, authors are suggested to specify the relative fold change values obtained from qPCR. Also, the fold change and p value of iron related genes filtered from dataset should be mentioned.
  • In figure 1 legends, authors should specify the technique used for analyzing the expression and whether the data is represented as mean with standard deviation or median with range. Legend 1E should be modified for more clarity.
  • In Figure 2, author should specify if the experiments were performed in duplicates or triplicates. If so, how are the values plotted. Also, if the statistical analysis is performed between the groups the level of significance (*) is only represented on one group. This should be reviewed.
  • Genes name should ne italicized.
  • In figures , the placement of * should be checked as statistical tests compare two groups. Legends should specify which test is being used to access the expression.
  • In section 3.5, authors are suggested to add the spearman rank values and p values of the significant correlations.
  • In figure 5, all the groups are not compared in Fig 5D-5E.
  • In 3.6, the route of injection should be specified,
Comments on the Quality of English Language

The quality of English should be improved so as to improve the clarity of the results. 

Reviewer 2 Report

Comments and Suggestions for Authors

Luo and colleagues have made a potentially important link between the paired homeodomain transcription factor PAX6, the hypoxia-inducible factor subunit alpha (HIF1a) and ferroptosis in glioblastoma cell lines.

1. Major concerns

Section 2.13 In Vivo Tumor Formation in Nude Mice and Section 3.6 In Vivo Validation (Figure 6):

a. There is no statement regarding animal care ethics institutional approval.

b. It is unclear why a group studying the impact of hypoxia on brain tumours would perform experiments by injecting glioma cells subcutaneously instead of orthotopically (i.e. intracranially). These techniques have been established for decades and the neuro-oncology community no longer considers the results of subcutaneous injection of tumour cells as valid for the study of brain tumours unless there is a comparison between subcutaneous and intracranial models in the same experiment. Moreover, how can the authors claim that their work can be translated to the clinic within insufficient in vivo evidence?

c. Figure 1, panel A. As presented (and unlike Figure 4, panel B), the results are not convincing regarding immunoreactive cells and their quantification in glioma patient derived FFPE sections. Further, within a region of interest, how were cells reproducibly counted (quantified)? Patient gliomas (grades 2, 3, 4) samples should be studied by Western blotting, not just the 2 glioma cell lines, to validate these results.

2. Specific concerns

a. Introduction

Line 94: Explain was "radicicol" is and its mechanism of action.

b. Materials and Methods

Line 121: What is different about the anti-PAX6 antibody (and the others) purchased from Cusabio? Are these monoclonal antibodies? What distinguishes this anti-PAX6 antibody from the one obtained from Millipore?

Line 129: Why were U251 and U373 glioma cell lines selected. for example U251 is p53 mutant. More details are required to justify their use.

Line 134: Immunohistochemistry: Negative controls (i.e. without primary antibody) for glioma tissue sections should be provided in the appendix/supplementary materials for the reviewers.

Line 144: There is no methods section for transient over-expression of cDNA constructs/plasmids (see lines 325 and 326 and Figure 2) into glioblastoma cell lines. It is sometimes unclear whether the authors are using transient overexpressed cells or stably expressed cells in their experiments. 

Line 224: What minimal RNA integrity number (RIN) was required for samples to proceed to RNAseq?

Line 268, Section 2.14 Immunohistochemical staining: How were IHC slides imaged and presented?

c. Results

Line 298: In the 5th edition of the WHO Central Nervous System tumour classification, tumour grades no longer use Roman numerals (i.e. I, II, III and IV); they have been replaced with standard numerals (i.e. 1, 2, 3, and 4).

Line 305-306: As a group, why were iron-metabolism related genes selected? Why were these four genes selected in particular? Spell out the full gene names at their first use in the manuscript.

Line 329: Where was Ferrostatin-1 (Fer-1) obtained commercially. How was the dosing determined? Please provide one or more references from the literature. Is the effect of Fer-1 dose dependent? If so, an additional dose of Fer-1 should be used in addition to the current dosing. In addition, a Pax6 knockdown experiment via siRNA is an important missing control for Figure 2.

Line 361 and Figure 3: Was the PAX6 over-expression from transient overexpression of a Pax6 plasmid or were these results from the stably transfected cells?

Line 415 and Figure 5. Panel H, third lane (Pax6OE + HIF1a-OE): Why is the band showing two parts?

Line 439: Based upon the use of subcutaneous tumour models, it is too preliminary to suggest that PAX6 could serve as a promising target for glioma therapy.

Line 441. Figure 6 legend. This figure uses both in vitro and in vivo studies; hence the title of the figure legend is incorrect to only use "in vitro".

d. The Discussion should propose important future directions to further unravel the mechanism whereby PAX6 regulates the expression of HIF1a. CUT&RUN using the Millipore antibody to PAX6 would be an important experiment to complete. Further, how would the authors propose to target a transcription factor such as PAX6 making it "druggable"? To date, only OLIG2 can be targeted, by a drug that prevents its dimerization.

3. Minor concerns

a. Abstract

Line 22: Add "The" prior to "paired". Replace "element" with "transcription factor".

Line 33: Replace "have been" with "were".

Line 35: Place a semi-colon (;) prior to "overexpression".

b. Introduction

Line 47: Rewrite as : "not eligible".

Line 54: Add "The" prior to "paired".

Line 68: Add 'and" after "glioma".

Line 70: Change "indicates" to "indicate".

Line 74: Replace "in" with "under" after "being".

Line 75: Rewrite as "promote cellular".

Line 83: Delete "role".

Line 91: Delete "the" prior to "ocular".

Line 100: Delete "support".

Line 101: Change "on" with 'in".

Line 114: Delete "the" prior to "ferroptosis".

c. Materials and Methods

Line 122: Delete one of the two alphas (α).

Line 195: Add "A" prior to "lipid".

Line 196 and 211: Explain ultrasonic "crushing". Most authors would use "sonication".

Line 203: Add "A" prior to "reactive".

Line 204: Spell out "DCFH-DA" in full at first use.

Line 210: Add "A" prior to "glutathione".

Line 219: What is meant by "gently blown"? Please use another term.

Line 271 and 281: What is meant by xylene "transparency". Most would write "xylene clearing".

Line 274: Replace "repair" with "retrieval" after "antigen".

Line 280: What is meant by "the blue was returned"? Please rewrite.

d. Results

Lines 352 and 356: Spell out all gene names at their first use in the manuscript.

Line 403: Delete "the" prior to "glioma".

Line 404: Change "induces" to "induced". Correct as "Additionally, further investigation...".

Line 438: Delete "the" after "that".

e. Discussion

Line 452: Delete the comma after "gliomas".

Line 462: Replace "As" with "Also".

Line 475: Delete "but" and "been".

Line 484: What is meant by "biofiducial"? Use another term understandable to most readers or explain the term.

Line 491: Delete "the" after "inhibiting".

Line 494: Add 'a" prior to "chronically".

Line 510: Change "proposed" to "proposes".

f. Author's contributions

Line 525: Change to "performed".

Line 527: Change to "handled".

Comments on the Quality of English Language

This reviewer has outlined numerous corrections under Minor Concerns.

Reviewer 3 Report

Comments and Suggestions for Authors

Luo et al. describe a set of observations suggesting a link between PAX6 and ferroptosis

in glioma cells via HIF1alpha. At first sight, the paper appears to include experiments building on each other. At second glance, the experiments using PAX6-overexpression do not convincingly support their statement that PAX6 is a regulator of ferroptosis. In some cases, experiments are even superficial, not well designed or described and need further controls. Thus, the paper needs major revision and new experiments before it can become accepted for publication.

Major flaws include that the author did not describe which PAX6 variant they employed for overexpression. They did not discuss the concentration-dependent transcriptional activity of PAX6 or which isoforms of PAX6 are expressed in healthy glial cells. The mentioned PAX6 primers are specific for zebrafish but by far not for humans. HIF1an is of course not the "N-terminal fragment of HIF-1α" (L399) but the Hypoxia inducible factor 1 subunit alpha inhibitor aka FIH. FIH hydroxylates HIF-1 alpha at Asn-803 in the C-terminal transactivation domain (CAD) preventing HIF1alpha interaction with Cbp/p300. In general, the resolution of the figures does not allow to reveal all the detailed information included. The figure legends are non-sufficient since they do not address always all items shown, nor do they not explain major statements. Moreover, they should contain statements regarding the statistics (method, number of experiments).

Figure 1:

First, Pax6 expression appeared downregulated in higher stages of gliomas. Looking at the pictures in A, the downregulation is not visible at least not obvious. The scales are not clear, nor the location of the magnification. The evaluation in B shows enormous error bars. It is unclear which protein (or if any) the authors address as internal reference. A controlled WB of the tissues shown in A should be included. The relative expression in C has been done with primers detected zebrafish pax6 but not human PAX6. Whether HFGC is a good control for a set of cancer glioma cells is not clear since HFGC is a fetal cell line, which may still express higher levels compared to adult glial cells. Thus, the impression of a downregulation in D may be an artefact of different stages of development/differentiation. E. Why is green control (what is the control?) and red glioma?

Figure 2:

A&B shows the accumulation of LPO upon PAX6-OE. The figure legend is misleading. LPO alone is an indication but not a prove that ferroptosis is induced. The authors have to include viability assays. C&D just shows that one inhibitor of ferroptosis can speed up cell proliferation again, but not that cell death did appear. Moreover, it is not clear what the control (NC) represents. Here, a transfection of a non-related transcription factor should be included to exclude that the observed effects are not due to simple massive overexpression of PAX6.

Figure 3:

How do the authors exclude that the effect of PAX6-OE is just a general stress response (see above). D is not a volcano blot. It is not stated whether the data in D-F is one cell line only or a combined analysis.

Figure 4:

ROS are not expressed. It is expected that in higher grade tumors HIF1alpha is upregulated due to hypoxia. The figure does not contain any data on PAX6-OE, see headline of the figure legend. Where is the relationship then? (see L373).

Figure 5:

It is not clear how the Spearman rank correlation analysis has been executed. The authors should be aware that Hif1an and HIF3a are considered as negative regulators of Hif1alpha/beta. No data on endogenous HIF1alpha is presented.

Figure 6:

In the present form, the result could be just a consequence of PAX6 overexpression slowing down cell proliferation, which simply diminish the need to build new vessel and therefore the expression of HIF1alpha.